# Network Biology Analyses and Dynamic Modeling of Gene Regulatory Networks under Drought Stress Reveal Major Transcriptional Regulators in *Arabidopsis*

**DOI:** 10.3390/ijms24087349

**Published:** 2023-04-16

**Authors:** Nilesh Kumar, Bharat K. Mishra, Jinbao Liu, Binoop Mohan, Doni Thingujam, Karolina M. Pajerowska-Mukhtar, M. Shahid Mukhtar

**Affiliations:** 1Department of Biology, 464 Campbell Hall, University of Alabama at Birmingham, 1300 University Boulevard, Birmingham, AL 35294, USA; nileshkr@uab.edu (N.K.); bharat26@uab.edu (B.K.M.); jinb2112@uab.edu (J.L.); binoopm2@uab.edu (B.M.); kmukhtar@uab.edu (K.M.P.-M.); 2Nutrition Obesity Research Center, University of Alabama at Birmingham, 1675 University Boulevard, Birmingham, AL 35294, USA; 3Department of Surgery, University of Alabama at Birmingham, 1808 7th Ave S, Birmingham, AL 35294, USA

**Keywords:** co-expression network, water deprivation, drought, systems biology, network centrality, computational simulation, *Arabidopsis*, transcriptional regulation

## Abstract

Drought is one of the most serious abiotic stressors in the environment, restricting agricultural production by reducing plant growth, development, and productivity. To investigate such a complex and multifaceted stressor and its effects on plants, a systems biology-based approach is necessitated, entailing the generation of co-expression networks, identification of high-priority transcription factors (TFs), dynamic mathematical modeling, and computational simulations. Here, we studied a high-resolution drought transcriptome of *Arabidopsis*. We identified distinct temporal transcriptional signatures and demonstrated the involvement of specific biological pathways. Generation of a large-scale co-expression network followed by network centrality analyses identified 117 TFs that possess critical properties of hubs, bottlenecks, and high clustering coefficient nodes. Dynamic transcriptional regulatory modeling of integrated TF targets and transcriptome datasets uncovered major transcriptional events during the course of drought stress. Mathematical transcriptional simulations allowed us to ascertain the activation status of major TFs, as well as the transcriptional intensity and amplitude of their target genes. Finally, we validated our predictions by providing experimental evidence of gene expression under drought stress for a set of four TFs and their major target genes using qRT-PCR. Taken together, we provided a systems-level perspective on the dynamic transcriptional regulation during drought stress in *Arabidopsis* and uncovered numerous novel TFs that could potentially be used in future genetic crop engineering programs.

## 1. Introduction

Since the last century, ecological and agricultural systems have been negatively impacted by a wide spectrum of environmental cues, including biotic and abiotic stresses. Among all the abiotic stimuli that limit crop yield, drought is the most threatening environmental constraint that adversely affects plant growth and productivity [1,2,3,4]. As such, it constitutes a serious threat to animals and crops in practically every corner of the globe by affecting up to 45 percent of the world’s agricultural area, which is home to 38 percent of the world’s human population [2]. In the United States, drought is increasingly common, costing an average of $9.6 billion each year, and is thus deemed as one of the most critical economic problems and natural threats [2]. Moreover, this crisis of crop productivity is further exacerbated by global climate change and future water shortages [3]. Since water is essential at all stages of growth and development, even a subtle variation in water potential during critical growth phases can adversely impact the yield of agronomically important crops [2,5,6]. For instance, drought-related water scarcity can reduce 40% and 50% of wheat and tomato production, respectively [5,7]. Likewise, it was projected that climate change-related water deficiency could reduce potato production by 18–32% in the next four decades [4]. Besides directly impacting crop yield, drought can induce nutrient deficiencies owing to a slower rate of mineral diffusion from the soil matrix to the roots [8]. Furthermore, drought stress elicits a wide range of physiological, biochemical, and metabolic injuries in plants, leading to cellular damage and a reduction in crop production and quality [2,9]. Therefore, it is imperative to understand the underlying molecular mechanisms of plant drought stress tolerance.

While diverse plant organs and tissues, including roots, shoots, and leaves, respond to drought stress in both a generalized and specialized manner [10], roots constitute the primary sensors detecting drought-related stresses. At the onset of drought stress, the modification of root architecture, as well as the regulation of root-to-shoot water traffic, are initiated to maximize water uptake [8,10,11]. To avoid non-stomatal water loss in the aerial parts of plants, drought induces structural and compositional cuticular modifications that augment the water barrier property [11]. Furthermore, stomata located in the epidermis of the leaves are closed to minimize excessive water loss through transpiration. Such physiological changes in the whole plant are coordinated with the alteration in the cellular concentration of diverse phytohormones, particularly the production of abscisic acid (ABA) and ethylene (ET) [10,12]. This leads to the activation of intricate hormonal signaling networks that are controlled by functional and regulatory proteins and manifested by global transcriptomic reprogramming [13,14,15]. Towards this, several major plant transcription factor (TF) families have been characterized in response to drought. This includes AREB/ABF (ABA-responsive element-binding protein/ABA-binding factor), DREB (dehydration-responsive element-binding), AP2/ERF (APETALA2/ethylene-responsive element-binding factor), bZIP (basic leucine zipper), NAC (NAM, ATAF1/2, CUC), MYB (myeloblastosis oncogene), and bHLH (basic helix-loop-helix proteins) [10,12,13,14,15,16,17]. While a wealth of drought-related transcriptomic information is available for the model plant *Arabidopsis*, it remains to be determined how temporal transcriptomic differences are governed by master regulators, including major TFs. To establish a comprehensive understanding of complex and dynamic host transcriptional regulatory responses to drought stress, a systems-level approach is necessitated that encompasses systematic, time-scale transcriptional response analyses in *Arabidopsis* that could be later applied to agronomically important crop plants.

A Systems Biology-based investigation of transcriptional regulatory networks (TRNs) entails the generation of genome-wide expression networks, identification of highly influential nodes, as well as dynamic mathematical modeling, computational simulations, and probabilistic assessments of transcription [18,19,20,21]. Regulatory networks comprise transcriptional components, referred to as “nodes,” and regulatory interactions between nodes, termed “edges” [20,22]. The characteristics of such transcriptional relationships constitute either direct TF-target interactions or co-expression of highly correlated genes [10,13,14,18]. Network biology, a branch of systems biology, is critical in deciphering the biological information of individual nodes or edges and regulatory networks as a whole [20,23,24,25]. Henceforth, topological features of a system and network architectural analyses can be employed to identify novel molecular players and regulators in various regulatory networks in diverse systems [14,26,27,28,29]. Among these network features, modules (clusters of highly correlated genes) and hubs (high degree; the number of edges of a node) have been used to predict major regulators in *Arabidopsis* and tomato [14,30]. Modeling dynamic transcriptional processes in TRNs is challenging, but it can hold great promise for comprehending how plants respond to water scarcity. For instance, in a small-scale genetics-based study, an *Arabidopsis* TF ANAC016 was shown to coordinate with AREB1 and NAP (NAC-like activated by AP3/PI) TFs through a trifurcate feed-forward regulatory circuitry in response to drought [31]. iDREM (interactive Dynamic Regulatory Events Miner) reconstructs dynamic regulatory networks at global levels that can potentially provide information on the function and timing of master regulators [32]. It utilizes time-course RNA-seq and static TF-target datasets and highlights major bifurcation events where the expression of a subset of genes diverges from that of the rest of the genes and identifies master regulators potentially responsible for the transcriptional events [32,33]. SQUAD (Standardized Qualitative Dynamical systems) perform steady-state studies and dynamic mathematical simulations of TFs and their interacting targets [34]. It combines Boolean and ordinary differential equation (ODE) models to execute such mathematical simulations [34]. Taken together, a system biology-based approach allows the construction of global drought-responsive TRNs and identifies master regulators that play critical roles throughout different stages of drought. 

Here, we identified time-point-specific unique and shared *Arabidopsis* transcriptome signatures in response to drought. Moreover, we highlighted the biological pathways pertinent to these temporal transcriptomes. We constructed a global co-expression network in *Arabidopsis* to determine highly significant modules that are implicated in drought. Our network architectural and centrality analyses identified high-priority genes and 117 TFs. Integration of TF-target and time-course RNA-seq datasets allowed us to identify 25 major TFs that are implicated in “water deprivation response” throughout the time of drought stress. We selected six of these novel TFs, which possess either of three centrality features, that is, hubs, bottleneck, and a high clustering coefficient (CC), and performed mathematical simulations that allowed us to decipher the amplitude and duration of target gene expression exerted by these six TFs. To support these computational transcriptional predictions, we provided experimental evidence of the TF-target relationship in a time-course drought qRT-PCR experiment using *Arabidopsis* plants. Overall, we provided a comprehensive dynamic model of the transcriptional regulatory network in *Arabidopsis* during drought stress and discovered several novel TFs that can be implicated in future genetic engineering crop enhancement programs. 

## 2. Results

### 2.1. The Temporal Transcriptional Landscape of Arabidopsis during Drought Stress Identifies Unique Transcriptome Signatures and Specific Biological Pathways

To reveal the drought-related transcriptional landscape in *Arabidopsis*, we built a systems biology-based pipeline (Appendix A). Towards this, we first analyzed a high-resolution mRNA expression dataset (GSE76827) that pertains to drought stress treatment on days 0, 1, 3, 5, 7, and 9 (Figure 1A, Appendix A). We observed only a few hundred differentially expressed genes (DEGs) in the early phase of drought stress i.e., 42 and 947 DEGs for day 1 and day 3, respectively (Figure 1B, Appendix A). However, we discovered a drastic change in gene expression as the drought stress progressed to day 5 (3924 DEGs) and day 7 (9285 DEGs), with the highest numbers of DEGs on day 9 (13,614 DEGs) (Figure 1B, Appendix A). Intriguingly, we found that similar numbers of genes were up- or down-regulated at all data points. For instance, 6702 and 6912 genes were up- and down-regulated on day 9 of drought stress, respectively (Figure 1B, Appendix A). It is important to note that the pattern of results is consistent with the original publication [35], but differs in number since we have chosen a different method, DESeq2, of analysis and cutoff for differential gene expression analysis. 

To uncover both the shared and distinct temporal transcriptional landscapes for each time point, we performed expression gradient analysis and identified time-point-specific unique DEGs. This method of gene expression analysis allowed us to determine numerous novel and unique transcriptional signatures over the course of drought progression. Specifically, we discovered 1, 46, 265, 940, and 5471 DEGs on day 1, day 3, day 5, day 7, and day 9, respectively (Figure 1C, Appendix A). These results indicate that the number of unique genes also gradually increased akin to the cumulative number of DEGs on a given day of drought stress. Compared to day 7, which exhibited only 940 unique transcriptional signatures out of 9285 DEGs, day 9 of drought stress is manifested with a significant number of unique DEGs, i.e., 5471 out of 13,614 DEGs (Figure 1C, Appendix A). Overall, these data suggest that while the amplitude and duration of transcriptional signatures might be different between day 1 and day 7 of drought stress, the transcriptional landscape is overall very similar for this phase of drought. Moreover, a significant transcriptional switch likely ensued on day 9 (Figure 1C, Appendix A). To further understand the biological characteristics of such transcriptional switches and gain insights into the potential pathways that are driving these time-point-specific DEGs under drought stress, we performed gene annotation and pathway analysis using Metascape [36]. We found that the “response to stimulus” gene ontology (GO) category was enriched on day 3, while day 5 was enriched with “pyrimidine nucleoside metabolic process,” “regulation of cell communication,” “sulfur and amino acid metabolic,” and “ion transport” GO categories (Appendix A). Intriguingly, day 7 GO categories were pertinent to cell wall biosynthesis, DNA damage, and cellular stress responses (Figure 1D). Finally, we observed that the drought response on day 9 featured GO categories related to catabolic processes such as “cell death” and “autophagy,” “proteolysis,” “vesicle-mediated transport.” “RNA splicing,” and “chromatin organization” (Figure 1E). Taken together, our analyses constructed a global, dynamic transcriptional landscape of drought that features major transcriptional events, unique gene signatures, and novel pathways implicated in the *Arabidopsis* drought responses. 

### 2.2. Network Topology Analyses of Drought-Related Co-Expression Network Reveal 117 Major TFs

A gene co-expression network is a systems biology-based analysis that isolates and highlights various correlation patterns among a set of genes and identifies diverse modules encompassing clusters of highly correlated genes that are potentially involved in related biological pathways. Such analysis can allow the prediction of gene functions as well as the discovery of functional modules, which may provide further insights into the transcriptional gene regulatory mechanisms. Towards generating an *Arabidopsis* Drought-specific Gene Co-expression Network (ADGCN), we used WGCNA (Weighted Gene Co-expression Network Analysis) [37]. Our ADGCN encompasses 9370 nodes (genes) connected by 402,598 weighted edges (co-expression relationship) (Figure 2A). To understand the nature of interactions within the network and ascertain highly connected modules in the ADGCN, we first calculated the degree of distribution and performed a connectivity analysis. We demonstrated that ADGCN exhibits the properties of a scale-free network (R2 > 0.9), a power-law degree distribution possessing fewer highly connected nodes with other nodes in a network (Appendix A). Subsequently, we demonstrated the network connectivity that was indicated by the location of genes in a dendrogram (Appendix A). Topological Overlap Mapping Metric (TOM) led to the identification of 15 diverse modules (Appendix A). Moving forward, we calculated the network features of ADGCN. We demonstrated that the average clustering coefficient and connectivity of ADGCN were 0.43 and 144, respectively (Appendix A). Subsequently, to retain the most contributing modules in ADGCN, we focused our analyses on five modules that displayed an increased clustering coefficient over the entire network. Furthermore, it is important to note that two of these five modules also possessed heightened connectivity over the entire network (Appendix A). Subsequently, we focused our analyses on five modules that displayed an increased clustering coefficient.. To provide functional insights into these modules, we performed pathway analyses (Figure 2A). The “blue” module was enriched with “amino acid homeostasis”, “autophagy”, “chemotaxis” and “mRNA splicing”. Interestingly, these GO categories were the hallmarks of unique DEGs of day 9 (Figure 2A). Likewise, we found that the significantly enriched GO terms of the “turquoise” module were “photosynthesis”, “biosynthetic process”, “mitotic cell cycle”, “DNA replication”, and “amino acid metabolism”, which were associated with the unique DEGs of day 5 and day 7 (Figure 2A). 

To identify the major TFs as well as additional transcriptional signatures that can significantly contribute to gene functions in the above-described molecular pathways, we next performed network architectural and centrality analyses. Since the topology of scale-free networks is largely controlled by highly connected nodes, we focused our analyses on the clustering coefficient, hubs (highly connected nodes), and bottlenecks (high betweenness nodes). Among the 592 TFs discovered in ADGCN, 51, 48, and 37 TFs exhibited high clustering coefficients, bottlenecks, and hub properties, respectively (Figure 2B–D and Appendix A). Given that 17 TFs were common between hubs and bottlenecks and only two TFs were shared between bottlenecks and clustering coefficients, cumulatively, we identified 117 non-redundant sets of high-priority TFs that possessed either of the three above-described centrality features. Major families that were enriched in this set of high-priority TFs included ANAC, activator protein 1 (AP1), bHLH, bZIP, Homeobox, MYB, nuclear factor Y (NF-Y), TCP domain protein (TCP), WRKY DNA-binding domain protein (WRKY), and Zinc finger (Appendix A). Finally, we also highlighted the functional GO categories of non-TF genes that were classified as high clustering coefficients, bottlenecks, and hub nodes. Notably, “alternative splicing”, “translational elongation”, and “polysaccharide biosynthetic process” gene function pathways were enriched in the high clustering coefficient nodes (Figure 2B,E). The “metabolism” and “amino acid” GO categories were exclusively enriched in bottlenecks (Figure 2C,E). Finally, we discovered that hubs were enriched in “cell cycle”, “DNA replication”, “photosynthesis”, “amino acid biosynthetic process” and “response to water deprivation” gene function classifications (Figure 2D,E). Overall, the generation of a comprehensive co-expression network and network topological analyses discovered potential major master regulators, nodes, and pathways in *Arabidopsis* drought signaling. 

### 2.3. Dynamic Transcriptional Regulatory Modeling Discovered Major Bifurcated Events of the TFs, Which Participate in the “Water Deprivation Response” Pathway

Direct transcriptional gene regulation relies on intricate coordination between TFs and their downstream target genes [32,33]. Therefore, inferring the dynamic regulatory relationships between TFs and their targets can provide a mechanistic view of how multiple TFs collaboratively participate in drought-related transcriptional circuitry during diverse stages of water deficiency stress. Towards this, we employed the iDREM package [32,33], which reconstructed drought-specific reaction networks in an unsupervised manner. Specifically, we integrated the publicly available time-course transcriptome data (GSE76827 with static TF-target datasets and performed dynamic regulatory modeling (see Section 4). We discovered significant bifurcation events in which the transcription of a subset of genes deviated from the expression levels of the remainder of the genes over the course of time, as well as identified all the TFs that might govern these sets of genes (Figure 3 and Appendix A). Furthermore, we also inferred the functional classification of the TFs and their targets based on the GO category data (Appendix A; *p*-value ≤ 0.05). Overall, we showed that 411, 323, 602, 212, and 546 TFs were modeled in diverse paths on days 1, 3, 5, 7, and 9, respectively (Figure 3, Appendix A). We subsequently focused our analyses on “response to water deprivation”, “response to stress”, “response to abiotic stimulus”, “response to chemical”, “response to hormone-mediated signal”, and “response to water” gene functional categories. Next, we directed our subsequent analyses to the “response to water deprivation” GO category. Towards this, we compared all the TFs in this category from our dynamic regulatory modeling with the ADGCN co-expression network and discovered a set of common 25 TFs (Appendix A). This set of selected TFs was enriched in bZIP, MYB, NAC, ERF, and AP1 superfamilies. In summary, our dynamic regulatory modeling allowed us to identify the transcriptional paths and major transcriptional events of previously undiscovered 25 TFs of “response to water deprivation” functional classification, which was also one of the categories enriched in the “blue” module of the ADGCN co-expression network (Figure 2A). The identification of these 25 critical TF regulators will substantially broaden our understanding of *Arabidopsis* drought-related transcriptional circuitry and provide candidate genes for downstream analyses in crops.

### 2.4. Mathematical Transcriptional Simulation of Six Major Tfs Acquired by Integrating Drought-Related Co-Expression Data with Dynamic Transcriptional Modeling

To simulate the influence of the regulatory activity of major TFs on their target genes, we used SQUAD simulations on an integrated network that included ADGCN and TF-target association data. For all the downstream simulations and experimental validations, we selected six major TFs that were present among the above-described 25 TFs of “response to water deprivation” and exhibited the properties of high clustering coefficient, hubs, and bottlenecks. This includes AT4G14770 (TESMIN/TSO1-LIKE CXC 2; TCX2/SOL2), AT1G51140 (ABA-RESPONSIVE KINASE SUBSTRATE 1; AKS1/FLOWERING BHLH 3; FBH3), AT5G56840 (a MYB-like TF), AT2G36270 (ABA INSENSITIVE 5; ABI5), AT3G09600 (LHY-CCA1-LIKE5; LCL5/REVEILLE 8; RVE8), and AT2G46590 (DOF AFFECTING GERMINATION 2; DAG2) (Figure 4 and Appendix A). Interestingly, our dynamic regulatory modeling indicated major bifurcation events for these TFs on day 3, day 5, and day 7 (Appendix A). We also showed that TCX2, AKS1/FBH3, AT5G56840, ABI5, LCL5/RVE8, and DAG2 co-regulate with 63, 146, 6, 121, 28, and 2 target genes, respectively. For mathematical simulations, the activation status of these six TFs was individually changed from 0 to 1 over the course of time to predict the dynamic behavior of their target genes. The transcriptional trajectory plots of the target genes illustrated varied levels of amplitude and intensity of activation for the corresponding putative transcriptional targets (Figure 4 and Appendix A). Overall, we simulated the transcriptional kinetics of target genes for a selected set of six major TFs to gain additional insights into their regulatory influence during the *Arabidopsis* drought response. 

### 2.5. Experimental Evidence for the Regulatory Relationship between Major Drought-Related TFs and Their Target Genes

To provide experimental validation of the dynamic gene regulation and transcriptional simulations, we performed extensive time-course qRT-PCR experiments in *Arabidopsis* wild-type Col-0 plants. We collected leaf samples of drought-stressed or corresponding well-watered control plants daily over a period of ten days. Based on the activation status, transcriptional trajectories, and amplitudes of gene regulation, we further selected four TFs (TCX2, AKS1/FBH3, ABI5, and LCL5/RVE8) and four targets for each of these TFs (Appendix A). TCX2 is a regulator of cell fate transition that controls divisions by regulating stem cell-type-specific networks [38]. We demonstrated that the mRNA levels of TCX2 were downregulated on day 2 and day 3, while its transcript levels were increased on day 5 and day 6 (Figure 5A). On the other hand, we observed a relatively negative transcriptional influence of this TF on its target gene, AT4G15790, particularly on day 1, day 5, and day 6 (Figure 5A). Interestingly, the other three tested target genes (AT4G23820; PGF13, Pectin lyase-like superfamily protein, AT5G38420; DEG24, a member of the Rubisco small subunit, and AT4G33680; AGD2, a diaminopimelate aminotransferase) positively co-expressed with TCX2 on day 5 and day 6 (Figure 5B–E; Appendix A). Moreover, TCX2 has also been shown to regulate stomatal cell lineages, further highlighting its connection to drought responses [38]. Our data indicate the potential roles of this stem-cell-expressed TF and its target genes in drought, particularly in the mid-phase of drought stress in guard cells. 

AKS1 (also known as FBH3) was shown to be released from the ABA signaling core complex, possibly in the guard cells [39]. Moreover, this basic helix-loop-helix type TF is also involved in photoperiodism flowering [39] (Appendix A). Our time-course expression data indicate that AKS1 is induced on day 5 of drought stress and its mRNA accumulation continues to increase for the remainder of the tested drought stress period (Figure 5F, Appendix A). While AT4G12560 (CPR1, a defense-responsive gene) positively co-expressed with AKS1 (Figure 5G, Appendix A), the transcript abundance of AT4G36740 (HB40, a homeodomain leucine zipper class I (HD-Zip I) protein) was negatively affected by AKS1 (Figure 5H). Moreover, we also showed that the expression levels of AT5G13330 (RAP2.6L, a member of the ERF) were suppressed during the later phase of drought stress (Figure 5I, Appendix A). Finally, the transcript levels of AT3G02310 (AGL4, a MADS-box protein) were only heightened on day 10 of drought stress (Figure 5J, Appendix A). These data further highlighted the complex involvement of the guard cells in various downstream responses to drought, and shed new light on how a master regulator controls additional TFs at various stages of drought stress. 

The third selected gene, AT2G36270 (ABI5), encodes a member of the basic leucine zipper TF family that is involved in ABA signaling during seed maturation and germination [40]. The transcript levels of this master regulator were increased on day 6, day 8, day 9, and day 10 (Figure 5K). Interestingly, two of its target genes (AT5G04760; DIV2 an R-R-type MYB protein, which plays negative roles in salt stress and is required for ABA signaling in *Arabidopsis*, and At4G08980; FBW2, a novel negative regulator of AGO1 protein levels that may play a role in ABA signaling) positively co-expressed with ABI5 (Figure 5L,M, Appendix A). On the other hand, the two additional tested targets of ABI5 (AT2G46680; HB-7 that encodes a putative TF in drought response, and AT4G33150; LKR that encodes a bifunctional polypeptide lysine-ketoglutarate reductase and saccharopine dehydrogenase involved in lysine degradation; Appendix A) were only induced during the very late phase of drought stress (Figure 5N,O). These data further underscore the complex and dynamic roles of ABA signaling at various stages of drought stress and may help provide a mechanistic understanding of how ABI5 controls global transcriptomic changes during drought. 

Finally, we examined the expression levels of AT3G09600 (LCL5/RVE8), which encodes an MYB-like TF. LCL5/REV8 was shown to be involved in the regulation of circadian signaling as well as the heat shock response [41]. We demonstrated that LCL5/RVE8 expression levels started to rise on day 5 and progressively increased until day 10 of drought stress (Figure 5P). Two of its targets (AT3G58630, a novel TF, and AT3G48990; AAE3, which encodes an oxalyl-CoA synthetase and is required for oxalate degradation) were positively co-expressed with LCL5/RVE8 (Figure 5Q,R Appendix A). However, the other two tested targets of LCL5/RVE8 (AT3G21890; BBX31, a B-box type zinc finger family protein, and AT4G34890; XDH1, which encodes a xanthine dehydrogenase that is involved in purine catabolism) were only regulated during day 5 and day 7 of drought stress (Figure 5S,T; Appendix A), further highlighting the LCL5/RVE8-mediated crosstalk between heat stress and drought. Therefore, LCL5/RVE8 could constitute a key regulatory node for genetic interventions in the face of increasingly common heat wave-drought events worldwide. 

Taken together, we experimentally characterized the transcriptional regulation patterns of four master TFs and their putative targets over a time course of drought stress in *Arabidopsis* plants. 

### 2.6. Experimental Evidence for the Regulatory Relationship between Significant Drought-Related TFs and Their Target Genes

To investigate the regulatory significance of the above-described TFs and their targets, we obtained mutants for FBH3, ABI5, and LCL5. We did not find any homozygous plants for TCX2 (Appendix A). Subsequently, we examined the expression levels of the respective TFs and their selected targets via qRT-PCR experiments under oxidative stress conditions simulated by methyl viologen (MV; N,-N′-dimethyl-4,-4′-bipyridinium dichloride; MV) (Figure 6 and Appendix A). Compared to the mock control, the mRNA accumulation of FBH3 was increased in MV-treated Col-0 seedlings. However, the basal and induced levels of FBH3 were dampened in the *fbh3* mutant. As alluded to above, the transcript levels of the target genes are possibly under the control of FBH3; we tested the expression levels of CPR1, HB40, RAP2.6L, and AGL4. Compared to the Col-0 control, both the basal and MV-induced levels of these four target genes were significantly reduced in *fbh3,* indicating the positive regulatory control of FBH3 on its target genes (Figure 6A–E). 

In the *abi5* mutant, ABI5 transcript levels were decreased compared to Col-0 in the mock condition. This reduction of the transcript was further exacerbated under MV-treated seedlings as Col-0 seedlings exhibited higher levels of ABI5. Subsequently, we revealed that the mRNA levels of all of its targets were reduced in the *abi5* mutant. Specifically, we demonstrated that the transcript level of one of its target genes, DIV2, decreased by up to two-fold in the *abi5* mutant under oxidative stress conditions. However, no significant change in the transcript level of DIV2 between the Col-0 and the *abi5* mutant was observed in the mock condition. Further, we observed a significant two-fold and four-fold decrease in the transcript level of FBW2 in the *abi5* mutant plant compared to Col-0 under both mock and treated conditions, respectively. The transcript level of HB-7 was more reduced in *abi5* mutant plants compared to Col-0 under both mock and stress conditions. We also observed a significant decrease in the transcript level of LKR in the *abi5* mutant under both mock and stress conditions in comparison to Col-0 (Figure 6F–J). 

In the *lcl5* mutant, we found that the level of mRNA transcript of LCL5 was ten-fold decreased than Col-0 in the mock condition. Moreover, under stress conditions, there was a significant reduction in the transcript level of LCL5 in the *lcl5* mutant plant. Similarly, we demonstrated that the transcript level of a novel TF (AT3G58630) was two-fold and three-fold decreased in the *lcl5* mutant compared to Col-0 under mock and oxidative stress conditions, respectively. Further, we observed a significant two-fold decrease in the transcript level of AAE3 in the *lcl5* mutant plant compared to Col-0 under stress conditions. The transcript level of BBX31 was also reduced in the *lcl5* mutant plants compared to Col-0 under stress conditions. However, no significant change in the transcript level of BBX31 between the Col-0 and *lcl5* mutant was observed in the mock condition. We also observed a substantial decrease in the transcript level of XDH1 in the *lcl5* mutant under both mock and stress conditions in comparison to Col-0 (Figure 6K–O). 

In summary, we provided genetic evidence for the regulatory relationships between major TFs and their targets in *Arabidopsis* plants. 

### 2.7. Anthocyanin Production in the Mutants Corresponding to Selected Drought-Related TFs in Response to Reactive Oxygen Species (ROS) and Oxidative STRESS-Inducing Agents 

Numerous reports have previously demonstrated an increase in ROS accumulation and oxidative stress in response to drought [42,43,44,45,46,47]. Specifically, we used Methyl viologen (MV; N,-N′-dimethyl-4,-4′-bipyridinium dichloride; MV) and paraquat (PQ), to induce ROS production. As described above, we did not find any mutant for the TCX2 gene; however, we included a mutant corresponding to DAG2. Since it has only two target genes, we did not perform any expression study but added phenotypic analyses. We subjected mutants corresponding to DAG2, FBH3, ABI5, and LCL5 to PQ and MV, followed by the quantification of anthocyanin accumulation to validate the physiological roles of the selected TFs. We observed negligible anthocyanin production in the mock (½ MS media) conditions. Interestingly, the level of anthocyanin was two-fold decreased in all the mutants, such as *fbh3*, *lcl5*, and *dag2,* compared to the Col-0 under MV treatment (Figure 7 and Appendix A). However, there was no significant variation in the level of anthocyanin production between *abi5* and Col-0. While in paraquat treatment, the anthocyanin content of all the mutants, including *fbh3*, *abi5*, and *lcl5*, decreased two-fold than that of Col-0. Moreover, in the *dag2* mutant, the amount of anthocyanin was two-and-a-half fold decreased compared to that in Col-0.

Taken together, we provided experimental evidence for the physiological roles of TFs under chemically induced oxidative stress conditions. 

## 3. Discussion

Drought is a primary abiotic environmental stress that limits agricultural productivity by impacting plant growth and development during all phases of their lives. Furthermore, global climate change and potential water constraints are likely to exacerbate the overall crop yield crisis [48]. Therefore, it is imperative to identify potential novel master regulators that can be used in genome engineering and editing programs to improve plant drought stress tolerance. Here, we used a systems-level approach to reveal phase-specific transcriptional signatures and identify biological pathways pertinent to the *Arabidopsis* drought transcriptome landscape. Our network science-based analyses unveiled major transcription factors that regulate drought response at various time points. Furthermore, dynamic regulatory modeling allowed us to uncover major regulatory events and the master regulators responsible for them. Finally, our mathematical transcriptional simulations followed by experimental validation provided potential molecular mechanisms on how master transcriptional regulators operate in an intricate regulatory network to respond to various phases of drought stress. 

In our global co-expression network analyses, we found 117 non-redundant sets of high-priority TFs that possess either of three centrality features i.e., clustering coefficient, hubs, and bottlenecks (Figure 2). Over the past decade, network analyses and centrality features have been widely applied to identify the most influential nodes in diverse networks, including protein-protein interactions and co-expression networks [14,18,26,27,28,29,49,50]. For instance, in a rather focused study, a protein phosphatase 2A (PP2A) network in *Arabidopsis* was generated [14]. Using the closeness centrality score (CCS), 13 PP2A genes were identified that were implicated in the synthesis of amino acids, nucleotides, fatty acids, phytohormones, and vitamin pathways [14]. Intriguingly, our phase-specific unique transcriptome signature analyses revealed that day 5 is enriched with “pyrimidine nucleoside metabolic process,” “regulation of cell communication,” “sulfur and amino acid metabolic,” and “ion transport” GO categories (Appendix A), suggesting the potential roles of PP2A in the mid-phase of drought stress in *Arabidopsis.* In another report, it was demonstrated that *Arabidopsis* PP2CA is also targeted by the bacterial pathogen *Pseudomonas syringae* to increase plant susceptibility, further supporting the complex role of PP2A genes in various plant stress responses [51]. 

In maize, over 800 hubs have been identified in a drought-associated co-expression network [52]. The pathway analysis of the non-TF hubs, glycolysis, and gluconeogenesis pathways were enriched, indicating that they might play an important role during leaf development. Moreover, carbon fixing, nitrogen metabolism, sulfur metabolism, linoleic acid metabolism, monoterpenoid biosynthesis, and glutathione metabolism were significantly altered during maize stress in drought [52]. While our 117 high-priority TFs were enriched in “alternative splicing,” “translational elongation,” and “polysaccharide biosynthetic process” gene function pathways (Figure 2B), our network topology analyses unveiled that the “metabolism” and “amino acid” GO categories were exclusively enriched in bottlenecks (Figure 2C). Finally, we discovered that hubs are enriched in “cell cycle,” “DNA replication,” “photosynthesis,” “amino acid biosynthetic process,” and “response to water deprivation” gene function classifications (Figure 2D). The major families of TFs that regulate the above-described pathways belong to ANAC, AP1, bHLH, bZIP, Homeobox, MYB, NF-YA, TCP, WRKY, and Zinc finger (Appendix A). Similar functional categories were found in the maize co-expression network, in which 49 hub TFs representing bHLH, bZIP, C2H2, Dof, MYB, ERF, and NAC were revealed [52]. Interestingly, 44 of these major TFs have also been shown to be involved in four distinct developmental stages of maize [52]. Likewise, the core component of the rice drought co-expression network is enriched in “sequence-specific DNA binding,” “porphyrin and chlorophyll metabolism” [53], which is potentially regulated by WRKY, Homeodomain, MYB, bZIP, Zinc finger domain, and C2H2 TFs. Overall, our data on *Arabidopsis* and other studies in monocot and dicot plants indicate the master regulators that are involved in drought-related biological pathways are conserved across diverse plant species. Given the involvement of these TFs in other abiotic and biotic stresses, as well as diverse plant development processes, it is plausible that plants rewire the flow of regulatory information and prioritize growth over survival responses during water deficiency conditions, similar to what was reported during biotic stress [54]. Finally, these datasets also point towards global crosstalk of diverse environmental stresses that require major nodes of convergence, possibly master transcriptional regulators. Therefore, the identification of such major transcriptional players is imperative to better inform future breeding programs. 

From the integration of the dynamic regulatory network and co-expression network analyses, we focused our mathematical transcriptional simulations and downstream analyses on a set of major TFs that represent “response to water deprivation”-related biological pathways (Figure 2, Figure 3 and Figure 4). Such an integrative systems biology-based framework can be expanded to all 117 major TFs, which exemplified other biological GO terms (Figure 2). It is important to note that previous studies in Arabidopsis or other models used co-expression network analyses and investigated the co-expression patterns in the entire module as a whole [55,56]. However, we integrated experimental TF-target datasets into our co-expression studies to understand the regulatory relationships, if any, across different modules. Moreover, previous studies have also used hubs within a module as an indicator of importance nodes [10,14,26,27,28,57,58]. Here, we focused on three diverse centrality measures, i.e., hubs, high betweenness, and high clustering coefficients. Our network science-based analysis also signifies the importance of our study to discover and characterize novel players in drought stress that have been implicated in response to diverse biotic and abiotic stresses, plant developmental processes, and phytohormonal-mediated crosstalk. One such master regulator that was discovered in this study is TCX2, a stem-cell-ubiquitous gene (Figure 5A). Interestingly, TCX2 is conserved in both plants and animals and belongs to the family of CHC proteins that are components of the DREAM cell-cycle regulatory complex [59]. As we had proposed, recent research has demonstrated that TCX2 functions as a master regulator, that orchestrates coordinated stem cell divisions through stem-cell-specific regulatory cascades in roots [38]. Interestingly, TCX2 and its closest homolog SOL1 regulate fate transition and cell divisions in the *Arabidopsis* stomatal lineage [60], suggesting a multifaceted role of this master regulator in roots and guard cells. While roots play a vital role in response to drought stress, it is also important to note that the regulation of guard cells to avoid water transpiration is equally significant. Thus, our data suggest a critical role of TCX2 in the early- and mid-phase of drought stress (Figure 5A. While TCX2 regulates a suite of diverse genes, we showed that it co-expresses and potentially regulates a pectin lyase (PGF13) [61], a Rubisco subunit (DEG24) [62], and a diaminopimelate aminotransferase, which is required for disease resistance to a bacterial pathogen [63]. Cumulatively, the evidence suggests the potential roles of TCX2 in photosynthesis, cell wall metabolism, and disease resistance. Notably, bacterial phytopathogens use stomata as the main route of entry into the leaf cells [64,65,66], further highlighting the roles of this major regulator in the crosstalk between biotic and abiotic stresses in plant developmental processes. 

Considering that stomatal closure is regulated by ABA to avoid further water loss during drought stress, it was unsurprising that two additional master regulators identified in the current study are major components of the ABA signaling pathway (Figure 5). AKS1 was shown to be released from the ABA signaling core complex, possibly in the guard cells, and connects the ABA core machinery and ABA-responsive genes [39]. Interestingly, our dynamic modeling, transcriptional simulations, and gene expression study further connected AKS1 to the phytohormone crosstalk, as it co-expresses and potentially regulates an Ethelene Responsive Factor (ERF) transcription factor, RAP2.6L (Figure 5I). While RAP2.6L was previously shown to be involved in resistance against a bacterial pathogen [67], AKS1 also potentially regulates CPR1 (a salicylic acid-dependent defense gene) (Figure 5G). Collectively, these data further underscore the importance of AKS1-dependent regulation of ABA-ET-SA crosstalk and, potentially, the control of stomatal aperture. ABI5 is the second major component of ABA signaling that is required at various stages of plant development, drought stress, and immune responses [40,51]. Among other targets, we experimentally demonstrated that ABI5 potentially regulates two ABA signaling players, DIV2 and FBW2 (Figure 5L,M), which negatively regulate ABA signaling [68,69]. Interestingly, we also showed the co-regulation of ABI5 with HB-7 (Figure 5N) and AKS1 with HB-40 (Figure 5H). Both of these homeodomain leucine zipper class I (HD-Zip I) possessed the properties of hubs and bottlenecks in our analyses, indicating the importance of these two key players in ADGCN (Appendix A). Remarkably, apple plants overexpressing HB-7 were more tolerant to drought treatment, maintained a high rate of photosynthesis, and accumulated increased biomass, suggesting the potential use of our network science-based discovery in genome editing and engineering programs for economically important plant species [70,71]. It is also important to point out that, while the current study examined the importance of transcriptome in discovering novel players in drought, this abiotic stress is manifested by intricate synergistic and antagonistic signals that cannot be fully captured by RNA-seq. Post-transcriptional and post-translation modifications, as well as 3D chromatin structure, lipidome, and metabolome, among other omics, needed to be considered when studying such a complex cellular process. 

In summary, our systems biology-fueled and network science-driven approach to modeling the dynamic transcriptional regulation in drought stress discovered novel master regulators that are involved in diverse biological pathways. Our integrative framework also highlights the intertwined relationships among phytohormone signaling networks, biotic/abiotic stresses, stomatal regulation/guard cell formation, and diverse plant developmental processes. 

## 4. Materials and Methods

### 4.1. Plant Growth Conditions

All work was performed with *Arabidopsis thaliana* ecotype Col-0. SALK_049022C (FBH3), CS873827 (ABI5), and CS840385 (LCL5) were purchased from the *Arabidopsis* Biological Resource Center (ABRC) and reconfirmed the mutant lines through PCR-based genotyping using TDNA insert and respective genes primers. Seeds were sown on Fafard Germination Mix in 90 mm × 68 mm Pöppelmann TEKU^®^ round pots, and cold stratified in the cold room at 4 ℃ for three days. Plants were grown at 23 °C with a 12 h light/12 h dark cycle with a light intensity of 120 µE/m2/s and watered every three days with tap water to keep soil moisture. Ten-day-old seedlings were transplanted to 12 × 6 Landmark Plastic flats with continuous irrigation until day 18. Half of the 18-day-old population was grown without further input of water, while the other half was supplied with 3 mL of tap water per plant 2 h prior to sampling. Samples were collected daily from day 18 to day 28 and flash-frozen in liquid nitrogen followed by storage at −80 ℃. Three leaves were harvested for each replicate, and three biological replicates were prepared. For mutant analyses, samples were harvested for PCR-based genotyping of the mutant plants after 7 days of growing into the flats. Two leaves from six individual plants for each mutant line were collected to extract DNA using Edwards’s buffer. Then, the extracted DNAs were used as the template to perform PCR using the Phire Hot Start II (Thermo Fisher Scientific, Waltham, MA, USA) enzyme. The amplified PCR products were analyzed through gel electrophoresis and documented under the Gel doc (BioRad, Hercules, CA, USA). The primer sequences are listed in the Appendix A. The confirmed mutant plants were further subjected to growth to collect seeds for the downstream experiments. 

### 4.2. Anthocyanin Quantification

*Arabidopsis* seeds were sterilized using a seed sterilization buffer containing ethanol, 10% Triton 10×, and sterile deionized (DI) water. The sterilized seeds were rinsed with sterile DI water 3–5 times. The sterile seeds were seeded on the ½ MS (Murashige and Skoog, PhyTotech labs, Lenexa, KS, USA) media, pH 5.8, containing square plates, and stratified for 3 days at 40C in the cold room. The plates were then transferred to the controlled growth chamber, maintaining a 12 h light/12 h dark cycle and 23 °C temperature conditions for germination. Five days post germination, seedlings of each mutant line, including wild type Col-0, were transplanted to the fresh ½ MS, 2 µM methyl viologen (MV, Thermo Fisher Scientific, Waltham, MA, USA) supplemented MS, and 2 µM paraquat (PQ, Thermo Fisher Scientific, Waltham, MA, USA) incorporated MS to simulate oxidative stress under controlled conditions. Seven days post-treatment, the plants were collected to perform anthocyanin measurements.

For anthocyanin quantification, 20 mg of fresh-weight leaves from each plant group were collected to extract anthocyanin using a solution containing methanol: hydrochloric acid: water in a 25:5:70 ratio. The whole seedling was incubated in anthocyanin extraction solution in the dark for 4 h, and the samples were centrifuged at 13,000 rpm for 5 min. The supernatant of 200 uL was aliquoted into 96 wells polystyrene plate (Thermo Fisher Scientific) to quantify the absorbance using a microplate reader (Aligent BioteK Synergy, Santa Clara, CA, USA) at 525 nm and 657 nm and estimated the anthocyanin content using this equation: anthocyanin content = (OD525 − 0.25 × OD657) [72]. For each group of treated and untreated conditions, 12 seedlings were used in each experiment. All the experiments were conducted with three biological replicates.

### 4.3. RNA Extractions and Quantitative Real-Time Polymerase Chain Reaction (qRT-PCR)

Leaf samples of Col-0 were ground with a Bead Ruptor 96-Well Plate Homogenizer. Total RNA was extracted using TRIzol (Invitrogen, Waltham, MA, USA) according to the manufacturer’s protocol. Briefly, 1 mL of TRIzol was used per sample. The RNA pellets were dissolved in 20 μL DEPC-treated water and quantified using BioPhotometer Plus (Eppendorf, Hamburg, Germany). 10 μg of RNA samples were then treated with DNAse using the TURBO DNA-free™ Kit (Ambion, Waltham, MA, USA) in a 20 μL reaction. cDNA was synthesized using SuperScript IV reverse transcriptase first-strand synthesis kit (Invitrogen) with 2 µg DNA-free RNA in a 10 μL reaction, as described previously [73]. PCR programs from both DNAse treatment and reverse transcription reactions were performed on an Applied Biosystems 96-Well Thermal Cycler (Thermo Fisher Scientific, Waltham, MA, USA). qRT-PCR was performed on an ABI 7500 Fast PCR System (Thermo Fisher Scientific, Waltham, MA, USA) with 2× PowerUp SYBR green master mix (Applied Biosystems, Thermo Fisher Scientific, Waltham, MA, USA) using the following settings: 50 °C for 2 min and 95 °C for 10 min followed by 40 cycles of 95 °C for 15 s, 55 °C for 15 s, and 72 °C for 1 min. Primer sequences are listed in Appendix A.

### 4.4. Drought Stress Analysis Data Acquisition

Drought-related data pertinent to this study were downloaded from the NCBI Gene Expression Omnibus (GEO) database [74]. Specifically, we employed the GSE76827 dataset that consists of transcriptome pertinent to drought-stressed aerial parts (shoot) of *Arabidopsis* at 0, 1, 3, 5, 7, and 9 days [35]. A custom Python script (version 3.8) was used to preprocess the sample data to filter out low-variance genes. During the preprocessing, genes that had zero expression values for more than 30 percent of the samples were pruned to maintain adequately high variance, which is necessary for the co-expression network construction. This approach provided data that were ready to proceed for downstream network-based and additional analyses. 

### 4.5. Differentially Expressed Genes (DEGs) Analysis and Functional Annotation Pathway Analyses

The processed transcriptome data from GSE76827 were subjected to iDEP [75], which is an interactive DEG (iDEP) analysis web tool, to compute DEGs between drought and mock treatments at diverse time points. Briefly, iDEP utilizes the DESeq2 R package to analyze DEGs [76]. We used a threshold of 1.5 fold change (FC), which is equivalent to 0.58 log2 Fold Change (log2FC), and a false discovery rate (FDR) ≤ 0.05 was set for DEGs [21]. DEGs were further subjected to hierarchical clustering. Functional annotation and pathway enrichment analyses were performed using Metascape using the cutoff 1.30 in -log10 scale (*p*-value ≤ 0.05) [36].

### 4.6. Weighted Gene Co-Expression Network Analysis (WGCNA) and Network Centrality Analyses

Towards generating an *Arabidopsis* Drought-specific Gene Co-expression Network (ADGCN), we utilized the GSE76827 dataset that consists of transcriptome pertinent to drought-stressed aerial parts of *Arabidopsis* at 0, 1, 3, 5, 7, and 9 days. We employed the WGCNA and (optional Limma) R packages, which provide a wide spectrum of R functions and carry out diverse weighted correlation network analyses. Specifically, this package includes functions for the construction of a co-expression network, detection of various modules with different correlation coefficients, analysis of a wide range of network topological features, simulation, and visualization of data and network [37]. WGCNA follows three major steps to build a co-expression network: (1) defining an adjacency matrix among all the genes based on pair-wise correlations, (2) calculating network centrality features from the adjacency matrix and converting it into a dissimilarity measure, and (3) application of hierarchical clustering by employing this dissimilarity measure. This is followed by tree cutting using either a static or a dynamic height cut. This analysis resulted in clusters or modules of genes with significance [37]. The data are formed into modules based on correlation patterns; colors are assigned to differentiate the modules. The nodes and edges data will be used to perform network analysis in the next step [49]. In the ADGCN co-expression network, the nodes correspond to genes, while weighted edges (≥0.75) define the interconnectedness based on expression correlations [77]. The edges-node data and module assignment data were further imported to visualize networks with module-assigned colors in Cytoscape [78]. After network construction, the topological features such as eigenvector centrality, degree centrality, betweenness centrality, closeness centrality, and information centrality are computed. After network construction, the topological features such as degree centrality, betweenness centrality, and clustering coefficient were computed using the Python-based NetworkX 2.6 package [79]. 

### 4.7. Dynamic Gene Regulatory Event Mining and Dynamic Simulations

To explore the dynamic behavior of water stress on *Arabidopsis,* we performed the Interactive Dynamic Regulatory Events Miner (iDREM) analysis on the drought-specific transcription response in *Arabidopsis*. The iDREM software employs static protein-DNA interaction data, including Chromatin Immunoprecipitation Sequencing (ChIP-Seq) ChIP-seq etc., and integrates with time-series gene expression data to simulate dynamic regulatory networks [32]. For the TF-target dataset, we curated TF and targeted gene regulatory networks by including vast datasets from the *Arabidopsis thaliana* Regulatory Network (AtRegNet) [80], Plant Cistrome Database (DAP_seq) [81], *Arabidopsis* transcriptional regulatory map (ARTM) [81], Curated_1 [82], TF2Network (Curated_2) [83], and Ath [80,82,83,84]. The *Arabidopsis* Information Resource (TAIR) [85], TF-gene interaction datasets, European Bioinformatics Institute (EBI)-based functional ontology [86], and drought-specific *Arabidopsis* expression data were used to perform iDREM analysis. iDREM highlighted major bifurcation events and displayed TFs that exhibited significant feature changes at a given time point. The score threshold was kept as 0.01, and significance was based on path significance conditional on a split, where the minimum split percentage was 50%. On each day of bifurcation, we highlighted the overall TFs expressed; afterward, we selected the top 100 TFs from the list based on a highly regulated quantity of genes and compared them with the *Arabidopsis* co-expression network. GO term categories were selected with a *p*-value ≤ 0.05. Finally, 6 TFs were selected from this group based on high network centrality features, including degree, betweenness, and clustering coefficient. Next, we focused on the “water deprivation” functional annotation. Mathematical simulations were performed using SQUAD (Standardized Qualitative Dynamical systems) [34], as described before [21]. Briefly, this analysis is based on standardized qualitative dynamical systems to simulate the behavior of regulatory networks from a steady (0) to an activated (1) state. Initially, at a steady state, the whole GRN (TFs and genes) have 0 values. To simulate dynamic behavior, The TF is activated from 0 to 1 only one time, which correspondingly modulates the dynamic behavior of genes in the regulatory network. In our Figure 4A–F, if the TF activation to 1 leads to changes in increased gene activity on the positive quadrant of the Y-axis, the TF may be predicted as an “activator.” However, if the TF activation to 1 leads to changes in increased gene activity in the negative quadrant of the Y-axis, the TF may be predicted as a “repressor.” Additionally, at the later time (pseudo-time) point, the decrease in activation indicates the loss of activity after the initial activation.

## Figures and Tables

**Figure 1 ijms-24-07349-f001:**
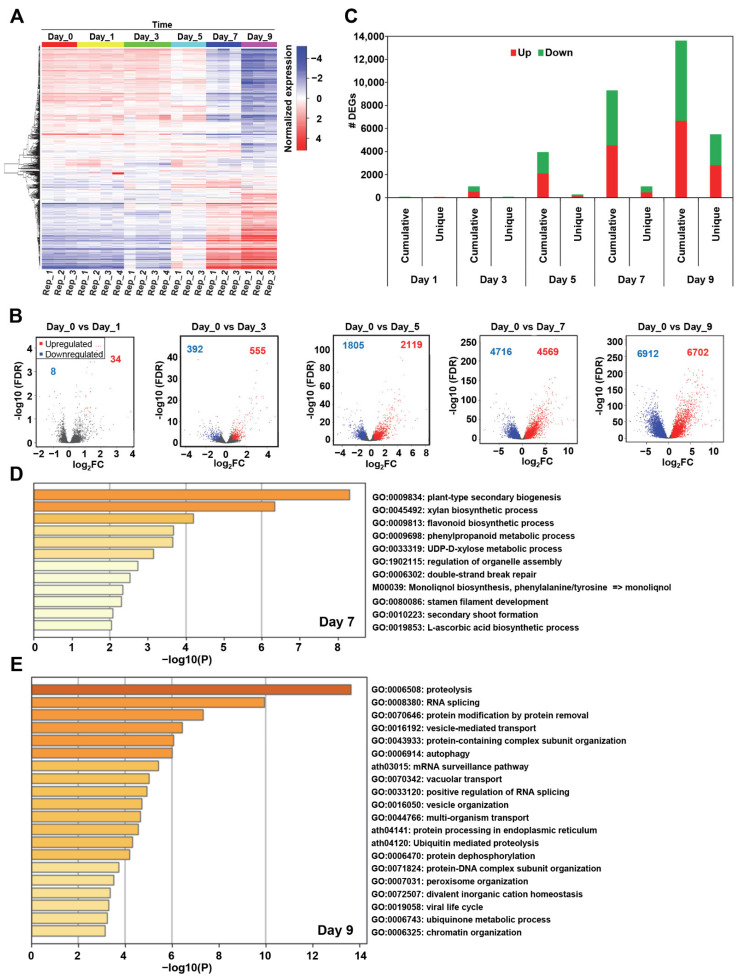
*Arabidopsis* transcriptome dataset (GSE76827) displays more change in the expression patterns at later sampling days of drought stress. (**A**) Heatmap of the gene expression profile of *Arabidopsis* under drought conditions on different sampling days (as indicated). The color bars display the normalized gene expression (Red ≥ 1, Blue ≤ −1). (**B**) The total number of uniquely expressed genes on days 1, 3, 5, 7, and 9 are shown in the plots. (**C**) Differential gene expression is displayed during the drought treatment time frame. The total number of genes significantly deferentially expressed were 42, 947, 3924, 9285, and 13,614, respectively, on days 1, 3, 5, 7, and 9(false discovery rate (FDR) ≤ 0.05, log2 Fold Change (log2FC) ≥ |0.58|). Red dots correspond to up-regulated genes, while blue points indicate down-regulated genes. (**D**,**E**) The gene ontology (GO) enrichment analysis of *Arabidopsis* unique differentially expressed genes (DEGs) on day 7; (**D**), and day 9; (**E**)) after drought stress is presented (*p*-value ≤ 0.05). The color gradient of bars in the gene enrichment analysis represents low (yellow) to high (red) enrichment significance values.

**Figure 2 ijms-24-07349-f002:**
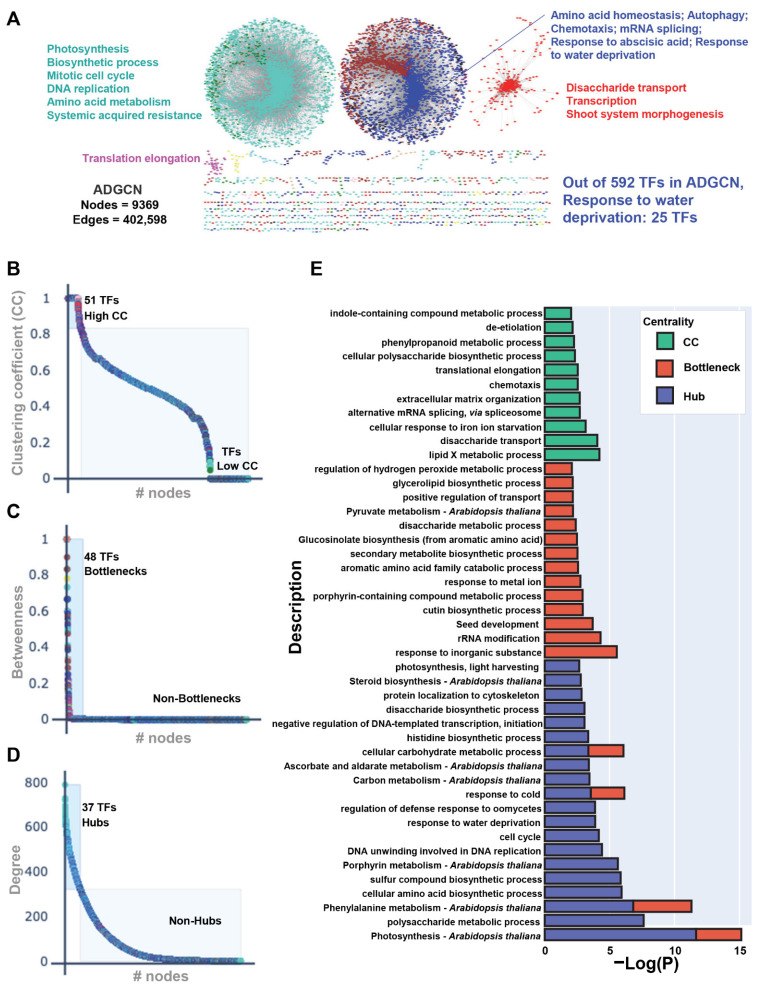
(**A**) Weighted gene co-expression network analysis (WGCNA) constructed the *Arabidopsis* Drought-specific Gene Co-expression Network (ADGCN) encompassing 9370 nodes connected by 402,598 weighted edges (≥0.75). Nodes are colored based on their specific module assignment. Most significant modules based on connectivity and clustering coefficient were annotated by GO analysis and the enriched pathways are listed next to their respective modules (*p*-value ≤ 0.05). (**B**–**D**) ADGCN network analysis plots displaying clustering coefficient (CC) (**B**), betweenness centrality (**C**), and degree distribution (**D**). (**E**) Functional annotation and pathway enrichment analyses of high CC (green), high bottleneck (red), and high hub (violet) genes.

**Figure 3 ijms-24-07349-f003:**
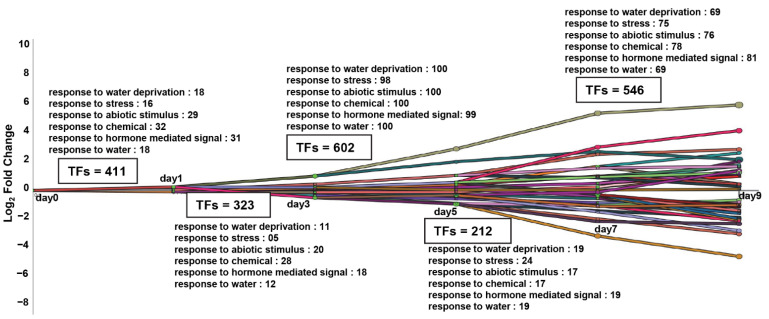
The plot illustrates the interactive Dynamic Regulatory Events Miner (iDREM) of drought response in *Arabidopsis* from day 0 to 9 at significance *p* < 0.01. The ontology identified the transcription factors (TFs) and genes involved in different functions on different days of drought stress. Each colored line corresponds to a unique transcriptional regulatory path and numbers of TFs representing each functional category are listed in text boxes.

**Figure 4 ijms-24-07349-f004:**
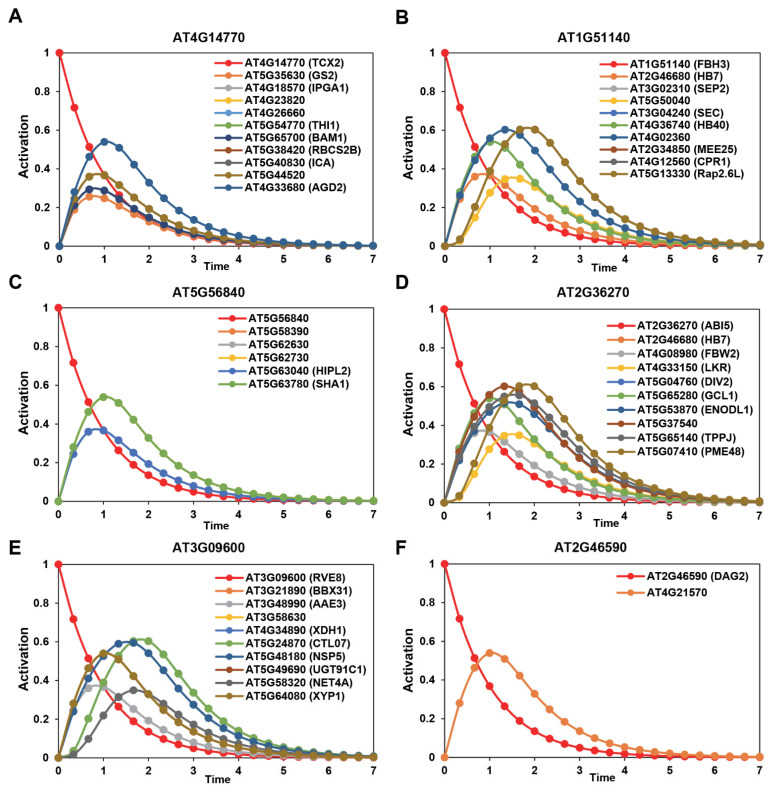
(**A**–**F**) The dynamic modeling and simulation of transcription factor regulatory network’s activity upon activation of six transcription factors (AT4G14770, AT1G51140, AT5G56840, AT2G36270, AT3G09600, and AT2G46590) identified by SQUAD. Activation patterns of the TFs are shown in red lines whereas individual target genes’ activation is illustrated in different colors.

**Figure 5 ijms-24-07349-f005:**
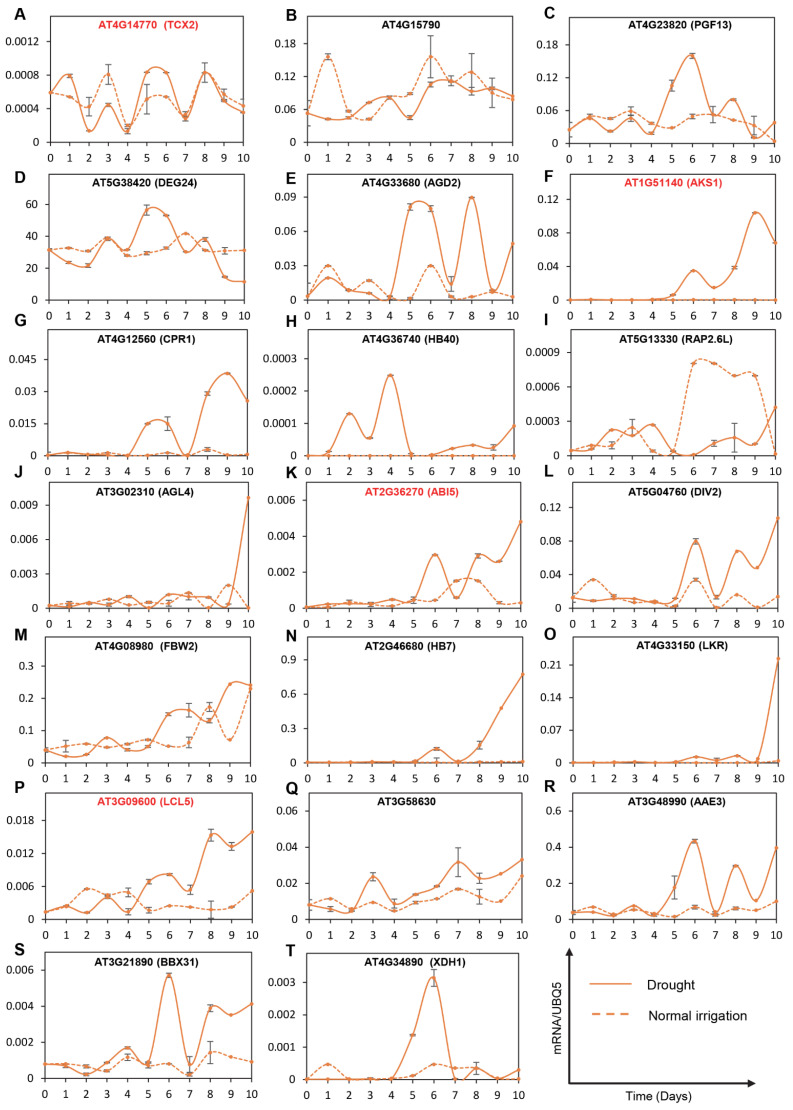
Kinetics of gene expression of transcription factors (red) and their targets in Col-0 under drought (solid lines) and normal irrigation (dotted lines) conditions. RT-qPCR was performed in leaf samples collected daily from day 0 to day 10. Gene expression was assessed using reference gene UBQ5 in four groups: TF1 (AAT4G14770 and targets (**B**) AT4G15790, (**C**) AT4G23820, (**D**) AT5G38420, (**E**) AT4G33680; TF2 (**F**) AT1G51140 and targets (**G**) AT4G12560, (**H**) AT4G36740, (**I**) AT5G13330, (**J**) AT3G02310; TF3 (**K**) AT2G36270 and targets (**L**) AT5G04760, (**M**) AT4G08980, (**N**) AT2G46680, (**O**) AT4G33150; TF4 (**P**) AT3G09600 and targets (**Q**) AT3G58630, (**R**) AT3G48990, (**S**) AT3G21890, (**T**) AT4G34890. The graphs represent the mean with standard errors of three technical replicates. Experiments were performed in three biological replications. The gray lines indicate error bars computed as the standard errors.

**Figure 6 ijms-24-07349-f006:**
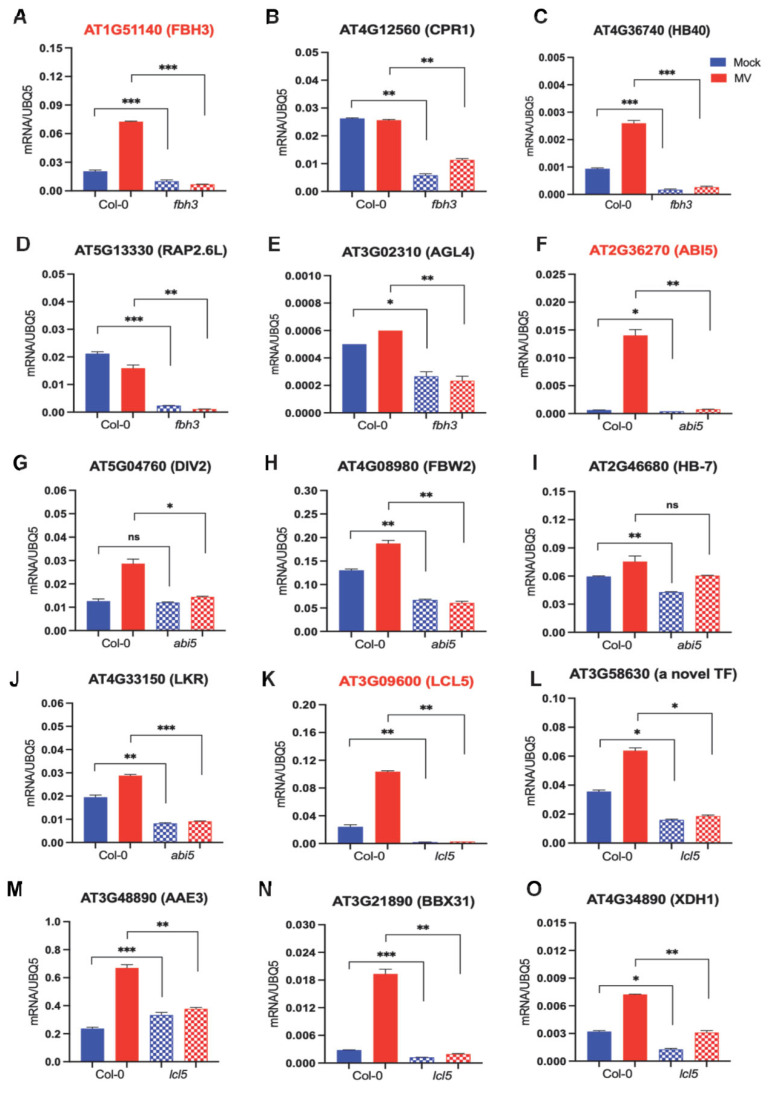
Gene expression profile of transcription factors (red) and their targets in Col-0, *fbh3* mutant, *abi5* mutant, and *lcl5* mutants in 1/2 MS media and 2uM methyl viologen (MV) treatment conditions. RT-qPCR was performed in leaf samples collected after 7 days of treatment. Gene expression was assessed using reference gene UBQ5 in three groups: TF1 (**A**) AT1G51140 and targets (**B**) AT4G12560, (**C**) AT4G36740, (**D**) AT5G13330, (**E**) AT3G02310; TF2 (**F**) AT2G36270 and targets (**G**) AT5G04760, (**H**) AT4G08980, (**I**) AT2G46680, (**J**) AT4G33150; TF3 (**K**) AT3G09600 and targets (**L**) AT3G58630, (**M**) AT3G48990, (**N**) AT3G21890, (**O**) AT4G34890. The graphs represent the mean with standard errors of three technical replicates. Experiments were performed in three biological replications. The statistical significance of the data is denoted by asterisks in the following manner: “* *p* < 0.05”, “** *p* < 0.005”, and “*** *p* < 0.0005”. The label “ns” indicates non-significant results, as determined by Student’s *t*-test. The lattice column indicates the significant differences between Col-0 and the mutants in comparison.

**Figure 7 ijms-24-07349-f007:**
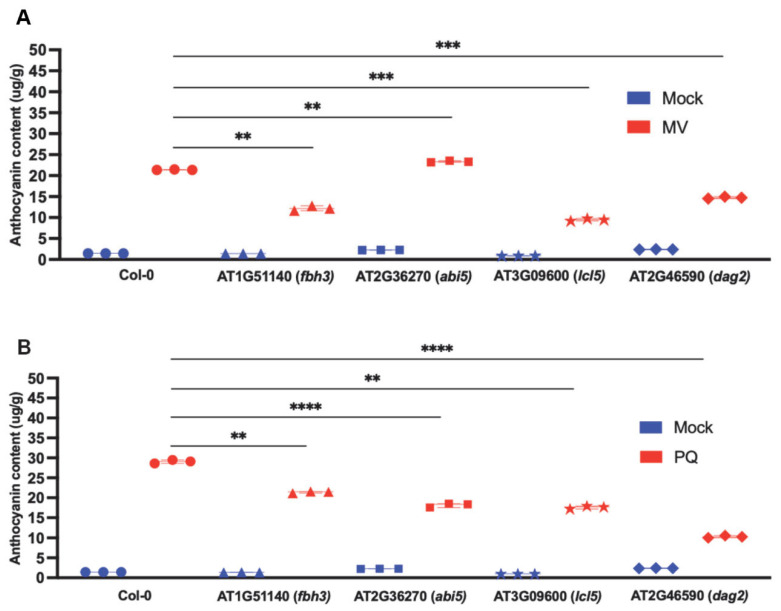
Anthocyanin content in leaves of 7 days old *Arabidopsis* plant in MS media and different stress conditions. (**A**) 2uM Methyl viologen (MV) treatment, *n* = 12 (three biological replicates) (**B**) 2uM paraquat (PQ) treatment, *n* = 12 (three biological replicates). Asterisks represented statistical significance (** *p* < 0.005, *** *p* < 0.0005, and **** *p* < 0.00005, Student’s *t*-test. The symbol ⬤ indicates Col-0, ▲ represents fbh3 mutant, ◼ signifies abi5 mutant, ★ identifies lcl5 mutant and ⯁ indicates dag2 mutant.

## Data Availability

The dataset analyzed in this study (GSE76827) can be found in the NCBI Gene Expression Omnibus (GEO) database (https://www.ncbi.nlm.nih.gov/geo/ (accessed on 4 August 2021)).

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
