# Peer review of "Network Biology Analyses and Dynamic Modeling of Gene Regulatory Networks under Drought Stress Reveal Major Transcriptional Regulators in Arabidopsis"

_ijms, 2023, doi:10.3390/ijms24087349_

Round 1

Reviewer 1 Report

This is an interesting manuscript on systems biology study of transcriptional regulation during drought stress in arabidopsis. The English style and grammar is fairly good, but needs to minor editing.

I have only few comments:

line 109-110: It seems that the sentence has an incorrect subject "a systems biology-based allows the..."
line 132: remove the dot after "this"
line 173: it seems that you need to change the sign in "Blue ≥ -1"
line 323, 326: repeated "Interestingly,..."
line 508: "Arabidopsis thaliana" should be italicized.

Author Response

line 109-110: It seems that the sentence has an incorrect subject "a systems biology-based allows the..."

we corrected the sentence as follow.

The actual lines stated by this question are line 115-117. The changes are made into “Taken together, the systems biology-based approach allows the construction of global drought-responsive TRNs and identifies master regulators that play critical roles throughout different stages of drought.”

line 132: remove the dot after "this".

            Removed this typo.

line 173: it seems that you need to change the sign in "Blue ≥ -1"

The actual line stated by this question is 369. Changes have been made from “Blue ≥ -1” to “Blue ≤ -1”. This line as stated by the reviewer is in the different lines (line 369 in figure 1 legend).

line 323, 326: repeated "Interestingly,..."

            we removed the repetition.

line 508: "Arabidopsis thaliana" should be italicized.

            The changes have been made into the main text.

Reviewer 2 Report

The present study focuses on establishing gene regulatory networks using a transcriptome dataset and predicting important TFs in gene regulation. The authors used a coexpression network analysis (WGCNA) to identify coexpression modules and important TFs and also used iDREM to understand dynamic regulatory networks in the same dataset. By intersecting the two resulting networks, the authors predicted 6 TFs and their potential targets. By performing qRT-PCR analyses, the authors confirmed the expression changes of 6 TFs and their targets under drought stress. In my opinion, this study is seemingly interesting but has several major issues for publication in this journal. 

Major issues:

1) WGCNA provides no regulatory relationship. It categorizes genes based on expression similarity. The high network scores do not mean a regulatory hierarchy. In other words, the selected TFs in the five modules found by WGCNA were pre-defined to be “regulators” in this paper, but they should not be. Fig 2B-D, the authors tested 592 TFs based on the three selected parameters, because of this reason. I do not agree with that. The authors should have investigated them among the entire module genes because there is no reason to exclusively look at TF genes in this network. WGCNA does not care what are TFs or not but quantifies “expression similarity” among all module genes. For this reason, I think that the approach taken in this paper is not reasonably acceptable. Even so, I am happy to hear opinions from authors. 

2) There are many existing studies of TF regulation in response to drought stress. The authors should provide an extensive on whether their prediction includes “marker TFs” in response to drought. For example, ABI5 is a key TF in ABA signaling and it is widely known to be regulated by posttranscriptional modifications (phosphorylation and ubiquitination) rather than transcriptional level. The approach author used heavily relies on only transcriptome data. 

3) Fig 5 (or Line 310): Fig 5 does not provide experimental evidence of the regulatory relationship. The authors confirm the gene expression of individual genes in a time of drought stress. I strongly suggest checking the expression of target genes in the corresponding TF gene knock-out mutant, which will confirm “the regulatory relationship”. ChIP-qPCR also needs to be performed to validate the vivo binding of at least 1-2 TF to the predicted target genes. Even though they used static TF-target data (e.g., DAP-seq), TF-target interactions are temporal-dynamic in vivo. 

Minor issues:

Line 133: cite the paper that GSE76826 was generated. Which tissue type (root or shoot) was used?

Line 136-140: explain if the result is consistent with one of the original paper. 

Line 150: I do not agree with “a majority”. 5471 out of 13614 is only “40%”.

Fig 1D-E: Provide information on the heat map (yellow-orange)

Line 189: Provide full information on each coexpression module and its eigengene

Fig 2A: Provide full information on GO analyses

Fig 2E: The blue text should be moved up to 1A (?)

Line 225 – 226: Provide reference(s)

Line 288: Provide details of how the 6 TFs were chosen (e.g., ranking %)

Fig 4C-F: According to the data, the activation level of all TF genes decreased from 0 to 1/2 during which one of all predicted targets increased. Does it mean that TFs are predicted to be repressors?

Fig. 5: I suggest labeling the TF panel with gene names for better visualization. Some panels are missing error bars (F, L, O…). 

Line 508: Italic for Arabidopsis thaliana.

Line 537: Deposit scripts to a public repository (e.g., GitHub) is strongly recommended if not mandatory by the journal policy. 

Line 4.5: What type of data was used for input? Normalized signal in the microarray in each condition or log2FC (stress/mock)?

Line 587-589: Provide details of the datasets. The current description is not sufficient. 

Author Response

We appreciate this reviewer for bringing up three main issues. We have extensively revised the manuscript in light of these comments. Moreover, we spent almost nine months providing genetics-based evidence about the regulatory relationships between the identified TFs and their targets (see new figures 6 and 7 as well as Figures S6 and S7). While details are discussed in the main point 3 (below), we obtained mutants for the corresponding TFs and tested the accumulation of mRNAs of the target genes in their respective tf mutant background. Moreover, we also provided evidence of these TFs in Anthocyanin accumulation in response to oxidative stress-inducing agents. 

Point-by-point answers to this reviewer's comments are detailed below.

Answer to the main issue 1: 

Firstly, we agree that WGCNA provides no direct information about regulatory relationships between TFs and targets. However, it is widely used to identify potential regulators of gene expression within specific modules or pathways, which was our aim in this study. We also agree that TFs and genes can be studied within certain modules. However, this does not narrow down the list of genes for downstream experimental evidence. Moreover, such studies have been done previously. 

It's important to note, however, that we used WGCNA in conjunction with iDREM as well as SQUAD. Moreover, we used gene ontology enrichment analysis and network centrality analysis, to identify potential regulatory factors that are enriched within specific modules or pathways identified by WGCNA. As alluded to in the manuscript, iDREM utilizes both the time-course dataset as well as existing TF-target relationships to infer regulatory relationships. As described in detail in the method section, we used several TF-target datasets. Specifically,  we curated TF and target gene regulatory network by including vast datasets from Arabidopsis thaliana Regulatory Network (AtRegNet) [75], Plant Cistrome Database (DAP_seq) [76], Arabidopsis transcriptional regulatory map (ARTM) [76], Curated_1 [77], TF2Network (Curated_2) [78] and Ath [75, 77-79]. The Arabidopsis Information Resource (TAIR) [80], TF-gene interaction datasets, the European Bioinformatics Institute (EBI)-based functional ontology [81], and drought-specific Arabidopsis expression data were used to perform iDREM analysis. 

Following such analyses, we were able to narrow down the potential TFs and their targets. Moreover, our new experimental data further provided solid evidence for these analyses. 

Finally, we would like to emphasize that our study aimed to identify potential regulatory TFs involved in drought stress response using an integrative approach. While we acknowledge that our approach has limitations, we believe that it provides valuable insights into the regulatory network underlying drought stress response and highlights potential novel targets for crop improvement strategies.

We have added a few sentences highlighting the importance of WGCNA as well as iDREM and SQUAD in the discussion section.

“It is important to note that previous studies in Arabidopsis or other models used co-expression network analyses and investigated the co-expression patterns in the entire module as a whole [54, 55]. However, we integrated experimental TF-target datasets into our co-expression studies to understand the regulatory relationships, if any, across different modules. Moreover, previous studies have also used hubs within a module as an indicator of importance nodes [10, 14, 26-28, 56, 57]. Here, we focused on three diverse centrality measures, i.e., hubs, high betweenness, and high clustering coefficients.”

Answer to the main issue 2:

As acknowledged in the main issue point 1, several previous have identified several regulators in drought and we have described them in the results section and thoroughly discussed them in the discussion section. However, in this study, we went on to highlight the importance of our work and provided details of newly identified players. 

We also agree on the importance of post-translational modifications as the reviewer stated in their comments. Likewise, post-transcriptional modifications as well as 3D chromatin structure and other -omics studies are equally important. However, the focus of this study is transcriptomics, which led to identifying novel players. Especially, the reviewer discussed the post-translational modifications of ABI5. It's important to note that its transcriptional regulation is equally important. 

We have added such limitations in the discussion section.

“While the current study examined the importance of transcriptome in discovering novel players in drought, this abiotic stress is manifested by intricate synergistic and antagonistic signals that can’t be measured by RNA-seq. Post-transcriptional and post-translation modifications as well as 3D chromatin structure, lipidome, and metabolome, among other omics, needed to be considered when studying such a complex cellular process.”

Answer to main issue 3:

To address this issue, we spent the last nine months in the wet lab and performed a series of new experiments. As suggested by the reviewer, we ordered knock-out mutants and genotyped them followed by testing the expression levels of the target genes in the respective tf mutant background under basal and induced conditions. We are happy to report the requirements of these TFs for the activation of all the tested target genes. Since the selection of the target genes was based on their co-expression as well as literature-based TF-target datasets, we believe that genetic experiments to display the regulatory relationships between TFs and their targets have been evidenced. Performing additional TF-target experiments are out of the scope of this manuscript. While the reviewer did not request, we further tested the phenotypic analyses of these mutants under oxidative stress conditions. Specifically, we showed the importance of Anthocyanin accumulation induced by two oxidative stress-inducing agents. We added two new main figures (Figs 6 and 7) and two new supplementary figs (Fig S6 and Fig. S7)

Minor issues:

We have addressed all of the minor issues in the manuscript.

We believe that we have addressed all the major and minor issues raised by this reviewer. We thank this reviewer for their critical evaluation as it has improved the quality of our manuscript.

Minor issues:

Line 133: cite the paper that GSE76826 was generated. Which tissue type (root or shoot) was used?

Citation as well as the tissue information have been added as follow.         

“Specifically, we employed the GSE76827 dataset that consists of transcriptome pertinent to drought-stressed aerial parts (shoot) of Arabidopsis at 0, 1, 3, 5, 7, and 9 days[ 35]”

Line 136-140: explain if the result is consistent with one of the original paper.

            The pattern of results is consistent with the original paper but differs in number since we have chosen different method DESeq2 of analysis and cutoff for differential gene expression analysis from the original paper.

We added the following sentence in the manuscript.

“It’s important to note that the pattern of results is consistent with the original publication [35], but differs in number since we have chosen different method DESeq2 of analysis and cutoff for differential gene expression analysis.

Line 150: I do not agree with “a majority”. 5471 out of 13614 is only “40%”.

            We changed it to “a significant number of ”

Fig 1D-E: Provide information on the heat map (yellow-orange)

            The following information is added in the Figure 1 legend.

“The color gradient of bars in the gene enrichment analysis represents low (yellow) to high (red) enrichment significance values. “

Line 189: Provide full information on each coexpression module and its eigengene

            Information is incorporated.

Fig 2A: Provide full information on GO analyses

            Information is incorporated.

Fig 2E: The blue text should be moved up to 1A (?)

            A change has been made into the main text. The new Figure is embedded.

Line 225 – 226: Provide reference(s)

            The authors are confused as line 225-226 are our own data and we have cited our own figures.

Line 288: Provide details of how the 6 TFs were chosen (e.g., ranking %)

            The following paragraph in section 2.4 reasoned the selection criteria.

For all the downstream simulations and experimental validations, we selected six major TFs that were present among the above-described 25 TFs of “response to water deprivation” and exhibited the properties of high clustering coefficient, hubs, and bottlenecks.

Fig 4C-F: According to the data, the activation level of all TF genes decreased from 0 to 1/2 during which one of all predicted targets increased. Does it mean that TFs are predicted to be repressors?

            We added the following information in the method section.

“Next, we focused on “water deprivation” functional annotation. Mathematical simulations were performed using SQUAD (Standardized Qualitative Dynamical systems) [34] as described before [21]. Briefly, this analysis is based on standardized qualitative dynamical systems to simulate the behavior of regulatory networks from a steady (0) to an activated (1) state. Initially, at a steady state, the whole GRN (TFs and genes) have 0 values. To simulate the dynamic behavior, The TF is activated from 0 to 1 only one time, which correspondingly modulates the dynamic behavior of genes in the regulatory network. In our Fig 4(A-F), if the TF activation to 1 lead to changes in increased gene activity on the positive quadrant of the Y-axis, the TF may be predicted as an "activator". However, if the TF activation to 1 lead to changes in increased gene activity on the negative quadrant of the Y-axis, the TF may be predicted as a "repressor". Additionally, in the later time (pseudo-time) point the decrease in the activation means the loss of activity after the initial activation.”

Fig. 5: I suggest labeling the TF panel with gene names for better visualization. Some panels are missing error bars (F, L, O…).

            The gene names have been included into the figure panels. The error bars were too small to be obvious in this figure, for which the necessary changes have been made to make them more visible. New Figure 5 is embedded.

Line 508: Italic for Arabidopsis thaliana.

            The change has been made into the main text.

Line 537: Deposit scripts to a public repository (e.g., GitHub) is strongly recommended if not mandatory by the journal policy.

            There is not any significant number of custom scripts used in this work that needs to be deposited into repository.

Line 4.5: What type of data was used for input? Normalized signal in the microarray in each condition or log2FC (stress/mock)?

            We downloaded the dataset and analyzed all by ourselves. The following information is provided in the method.

The processed transcriptome data from GSE76827 was subjected to iDEP[75], which is an interactive DEG (iDEP) analysis web tool, to compute DEGs between drought and mock treatments at diverse time points. Briefly, iDEP utilizes DESeq2 R package to analyze DEGs [76].  We used a threshold of 1.5 fold change (FC) which is equivalent to 0.58 log2 Fold Change (log2FC) and a false discovery rate (FDR) ≤ 0.05 was set for DEGs [21]. DEGs were further subjected to hierarchical clustering. Functional annotation and pathway enrichment analyses were performed using Metascape using the cutoff 1.30 in -log10 scale (p-value ≤ 0.05) [36].

Line 587-589: Provide details of the datasets. The current description is not sufficient.

            Information is provided for qRT-PCR. All primer information is present in Table S3.

Round 2

Reviewer 2 Report

In revising their manuscript, the authors addressed all issues raised before, carefully responded to my comments, and included additional information.